# Psychological interventions for weight reduction and sustained weight reduction in adults with overweight and obesity: a scoping review protocol

Oliver Hamer ,[1] Emma P Bray ,[1] Cath Harris,[1] Amy Blundell,[1] Jennifer A Kuroski,[1] Emma Schneider,[2] Caroline Watkins,[1] Andrew Clegg[1]

¹University of Central Lancashire, Preston, UK
²Liverpool University Hospitals NHS Foundation Trust, Liverpool, UK

**Correspondence to**
Dr Oliver Hamer;
OHamer@uclan.ac.uk

## ABSTRACT

**Introduction** Overweight and obesity are growing public health problems worldwide. Both diet and physical activity have been the primary interventions for weight reduction over the past decade. With increasing rates of overweight and obesity, it is evident that a primary focus on diet and exercise has not resulted in sustained obesity reduction within the global population. There is now a case to explore other weight management strategies, focusing on psychological factors that may underpin overweight and obesity. Psychological therapy interventions are gaining recognition for their effectiveness in addressing underlying emotional factors and promoting weight loss. However, there is a dearth of literature that has mapped the types of psychological interventions and the characteristics of these interventions as a means of achieving weight reduction and sustained weight reduction in adults with overweight or obesity.

**Methods and analysis** The review will combine the methodology outlined by Arksey and O'Malley with the Preferred Reporting Items for Systematic Reviews and Meta-Analyses extension for Scoping Reviews guidelines. A total of six databases will be searched using a comprehensive search strategy. Intervention studies will be included if participants are 18 years and over, classified as overweight or obese (body mass index ≥25 kg/m²), and have received a psychological therapy intervention. The review will exclude studies that are not available in English, not full text, none peer reviewed or combine a lifestyle and/or pharmacological intervention with a psychological intervention. Data will be synthesised using a narrative synthesis approach.

**Ethics and dissemination** Ethical approval is not required to conduct this scoping review. The findings will be disseminated through journal publication(s), social media and a lay summary for key stakeholders.

## STRENGTHS AND LIMITATIONS OF THIS STUDY

⇒ A comprehensive literature search of six electronic databases has been designed collaboratively with a skilled information specialist.
⇒ The protocol for the review has been codeveloped with a public advisor who has over 10 years' experience living with obesity.
⇒ The scoping review will be limited to comparisons with other psychological interventions, usual care or no intervention and will not, therefore, include a comparison with other active lifestyle interventions (unless classified as usual care).
⇒ Relevant papers written in other languages could be omitted from this review, given the inclusion criteria to only include papers written in English.

## INTRODUCTION

Obesity is a growing public health problem worldwide.[1] It has recently been recognised (and classified) by The Obesity Society and the European Commission as a chronic disease.[2–5] Globally, approximately 650 million adults and 340 million children and adolescents (ages 5–19) currently suffer from obesity.[6] Obesity is known to increase the risks of a wide spectrum of comorbidities, including cardiovascular disease, type 2 diabetes, cancer, hypertension and several types of musculoskeletal disorders.[7–9] It has a multifaceted aetiology, with factors such as genetics, medical conditions, stress and environmental factors contributing to the prevalence of obesity.[1]

Energy balance (energy input from food consumption and energy expenditure from physical activity) is one of the key factors to weight maintenance and, hence, diet (energy intake) and physical activity levels (energy output) are key behaviour determinants of obesity.[10] Over the last 20 years, diet and physical activity have been the primary interventions recommended for weight loss.[11] However, despite the efficacy of these interventions, long-term weight loss maintenance remains a challenge, with many individuals regaining lost weight.[12] With worldwide prevalence of overweight and obesity rates continuing to rise, it is evident that dietary and physical activity interventions have not resulted in a sustained obesity reduction

within the global population.[13] The lack of longer-term (or population) success associated with these interventions may be due to a primary focus on increasing energy expenditure and reducing dietary intake.[14] While these factors are critical for weight loss, they ignore the complex psychological factors (eg, emotional, such as fears and behavioural, such as avoidance) that underpin overweight and obesity.[6 15–18] These factors can contribute to disordered eating, low self-esteem, depression, anxiety and stress (which may lead to weight gain).[15 19] Furthermore, these interventions often ignore the environmental factors that contribute to weight gain, such as food accessibility, social support and cultural norms.[20] With the previous emphasis on diet and physical activity interventions seemingly having little impact on population prevalence of overweight and obesity, there is now a case to explore additional weight management strategies, focusing on the psychological factors that may underpin overweight and obesity.[17 21]

In recent years, there has been increasing recognition of the importance of psychological interventions for weight reduction in adults with overweight and obesity.[22] Psychological interventions address the underlying psychological and behavioural factors that contribute to excessive weight.[23] These interventions support individuals to develop skills and strategies to cope with stress, negative emotions, and cognitions, modify unhelpful behaviours, and promote healthy lifestyle changes.[22 23] They also aim to increase self-efficacy, self-regulation and self-awareness, which are critical for long-term weight reduction and maintenance.[24] By targeting the psychological and behavioural factors that contribute to overweight and obesity, these interventions may be more effective for long-term weight loss maintenance compared with diet and physical activity interventions alone.[25] These interventions may be delivered individually or in groups, in clinical, community, workplace and online settings, and include a variety of components such as cognitive behaviour change, mindfulness, relaxation and motivation.[23 24]

As research in the field of weight reduction and management continues to evolve, it is important to identify novel interventions.[26] This is particularly important for psychological interventions, which are often overlooked in favour of more traditional approaches such as diet and exercise.[24] Several systematic reviews have been conducted to assess the effectiveness of psychological interventions for weight loss in overweight or obese adults.[23 24] However, these reviews are limited by population (including only adults with specific medical conditions), inclusion of interventions (including only one type of psychological therapy intervention) or being substantially outdated.[23 24 27] With the rapid development of new technology (eg, digital health technologies) and therapeutic techniques, coupled with the ever-increasing rates of overweight and obesity, a scoping review is now needed to map the different types of psychological interventions and their core characteristics, and identify

evidence gaps and areas for future research in this field.[28]

## Aim
To explore psychological interventions for adults with overweight or obesity as a means of achieving sustained weight reduction.

## Objectives
1. To generate a comprehensive map of the types of psychological interventions for weight reduction and sustained weight reduction in adults with overweight or obesity.
2. To identify the core intervention characteristics including theoretical foundations, techniques, setting, duration, mode, number of sessions and provider of psychological interventions for weight reduction and sustained weight reduction in adults with overweight or obesity.
3. To identify study characteristics such as length of follow-up, and outcomes of studies which include psychological interventions for weight reduction and sustained weight reduction in adults with overweight or obesity.
4. To generate a comprehensive map of the types of usual care for weight reduction and sustained weight reduction in adults with overweight or obesity.
5. To identify the core characteristics including components, setting, duration, mode and providers of usual care for weight reduction and sustained weight reduction in adults with overweight or obesity.
6. To identify evidence gaps and areas for future research in the field of psychological interventions for weight reduction and sustained weight reduction in adults with overweight and obesity.

## METHODS AND ANALYSIS
### Study design
The study will follow a scoping review design, following the framework set out by Arksey and O'Malley, which recommends a five-stage review process[29]:
► Stage 1. Identifying the research question.
► Stage 2. Identifying relevant studies.
► Stage 3. Study selection.
► Stage 4. Charting the data.
► Stage 5. Collating, summarising and reporting the results.

We will not implement the optional sixth stage set out by Arksey and O'Malley as the first five stages are adequate to satisfy the reviews aims and objectives.[21] The study will be reported in accordance with the Preferred Reporting Items for Systematic Reviews and Meta-Analyses extension for Scoping Reviews (PRISMA-ScR) guidelines.[30]

A scoping review design was chosen because it enables an exploration of the broader spectrum of evidence relating to the range of psychological therapy interventions for adults with overweight and obesity, without being constrained by a rigid inclusion and exclusion criteria of a

systematic review.[28] Given the absence of previous reviews which have explored the objectives of this study, the scoping review methodology allows for a more expansive inclusion criteria (eg, different study designs and intervention types) which will ensure that all relevant studies will be included. The scoping review design has been found to be well-suited for topics that are rapidly evolving and include a heterogeneous body of literature, which is applicable to psychological interventions as a means for achieving weight reduction and sustained weight reduction (in adults with overweight and obesity).[28]

### Search strategy

Six databases will be searched to identify relevant articles:
1. Cochrane Central Register of Controlled Trials (CENTRAL).
2. World Health Organization International Clinical Trials Registry Platform (ICTRP).
3. EMBASE.
4. MEDLINE.
5. CINAHL.
6. PsycINFO.

We will search the above databases and trials registries from their inception to the present, with no limiters related to year of publication. A search of clinical trials will be conducted through the CENTRAL and ICTRP databases to identify ongoing clinical trials. We will check the reference lists of all included studies and relevant review articles for additional studies.

The search strategy conducted in each database will be based on the following:
► Population: Adults with overweight or obesity (aged ≥18, with BMI≥25 kg/m$^2$).
► Intervention: Psychological interventions (eg, cognitive–behavioural therapy, mindfulness, motivational interviewing).
► Study type: Randomised controlled trials (RCTs) or non-randomised comparative intervention studies.

The search strategy, designed by the research team in collaboration with an expert information specialist, will be adapted for each database (see online supplemental appendix A for Ovid example). The search strategy was based on the strategy outlined in the Cochrane review by Shaw *et al*,[24] but with the addition of further search terms identified by the research team, and the Canada's drug and health technology research working group, CADTH search filter (to identify randomised and non-RCTs).[24 31]

### Study selection

All articles identified by the electronic database searches will be imported into Rayyan (web application). One reviewer will then independently screen the titles and abstracts of the search results for appropriate studies. Following initial screening, two reviewers will independently screen the full texts of all potentially eligible studies. The two reviewers will resolve disagreements through discussion with a third reviewer. Full-text articles that meet the inclusion criteria will be charted to summarise the findings. The study selection process will be recorded in sufficient detail using a PRISMA flow diagram, reporting reasons for exclusion.

### Data extraction

One reviewer will independently extract data for the included studies using a piloted data extraction form. A second reviewer will check and verify all the extracted data. Data will be extracted on the following: study methods, participants, interventions (eg, type, setting, components), outcomes and any other information judged to be important to meet the objectives of the study.

### Inclusion criteria

While a scoping review traditionally encompasses a wide range of literature, specific criteria will be employed to identify relevant articles for the aim and objectives of the study.

### Types of studies

RCTs and non-randomised comparative intervention studies will be included. We will only include studies reported in full text. Whereby we identify an abstract meeting the inclusion criteria, we will contact the study authors to establish if a full text is available.

### Types of participants

Studies will be included if the participants are ≥18 years and were classified as overweight or obese (body mass index, BMI≥25 kg/m$^2$). Within studies, BMI, or the components of BMI, may be self-reported or measured. Studies will also be included when a sample includes participants with other classifications of BMI (eg, underweight, healthy weight) if data relating to those with overweight or obesity can be extracted separately.

Studies will be included whereby participants with overweight or obesity have other co-morbidities (eg, musculoskeletal pain, osteoarthritis, diabetes, hypertension, arthritis, hyperuricaemia, gall bladder disease), as BMI>25 kg/m$^2$ has been associated with the incidence of multiple comorbidities.[32]

### Types of interventions

Psychological therapy interventions will be defined as those interventions that involve meeting with a therapist (healthcare professional competent in providing psychological therapy) to discuss feelings and thoughts and how these affect behaviour and well-being.[33] All psychological interventions will be considered for inclusion. Some of these interventions include cognitive–behavioural therapy (CBT), mindfulness, psychodynamic therapies, motivational interviewing (MI), prolonged exposure therapy, hypnotherapy, relaxation, psychotherapy, rationale emotional therapy and systematic desensitisation. Individual and group therapies will be included.

We will only include studies of psychological therapies that are designed to reduce weight or increase health promoting behaviour for the purpose of sustained weight reduction (ie, eating behaviour or physical activity).

To identify psychological interventions for weight reduction and sustained weight reduction, we will include studies with the following comparisons:

► Psychological therapy versus no intervention.
► Psychological therapy versus usual care (as defined by study authors).
► Comparisons between different types of psychological therapy (eg, CBT, mindfulness, MI).

We will not compare psychological interventions with other active interventions (eg, diet and exercise interventions), unless these are classified as usual care.

## Outcomes

Information on the following study outcomes, where available, will be collected: body weight or indicator of body mass (eg, BMI), adverse events, quality of life, physical activity levels and eating behaviours (eg, constructs of dysfunctional, uncontrolled, restrained, emotional and adaptive eating behaviours as measured by instruments such as the Emotional Eating Scale, Intuitive Eating Scale-2, Mindfulness Eating Questionnaire, Dutch Eating Behaviour Questionnaire, Three-Factor Eating Questionnaire)[34–38]

## Exclusion criteria

We will exclude studies that are not available in English, are not full text articles, and not peer reviewed articles (eg, magazine articles, letters, editorials, newspaper and commentary articles).

## Data synthesis and analysis

A narrative synthesis will be employed to synthesise the data, prioritising and ordering data using the guidelines of the Synthesis WIthout Meta-analysis in systematic reviews.[39]

## Patient and public involvement

A public advisor with more than 10 years' experience of living with obesity has been involved in designing the review and this protocol paper, and will continue to have input throughout the project, including the determination of the objectives and improving the lay interpretability of the review. The public advisor will also help to codevelop the dissemination strategy.

## What the findings may add

This scoping review is likely to identify specific gaps within existing evidence that may be useful in shaping future research initiatives. The identification of these gaps could pave the way for the development of novel innovative psychological interventions and may provide a roadmap for researchers to explore the effectiveness of these interventions. The findings of this review may also assist healthcare providers and key public health organisations gain insight into the types of psychological interventions that can be utilised as a means of weight reduction and sustained weight reduction for adults with overweight and obesity. With a greater understanding of psychological interventions (ie, the types, characteristics and impact) for weight reduction and sustaining weight reduction, key organisations and healthcare services can tailor programmes to include evidenced based psychological components.

## Ethics and dissemination

Ethical approval is not required to conduct this scoping review. The findings will be submitted to a peer-reviewed journal for publication. A lay summary will also be prepared with the assistance of the patient and public involvement author and disseminated on social media and to other key stakeholders involved in the care of adults with weight concerns. Findings will also be shared on the National Institute for Health and Care Research, Applied Research Collaboration - Northwest Coast (NIHR ARC-NWC) social media pages.

**Contributors** OH was responsible for the conception, design and writing of the manuscript. CW, EPB and AC provided support throughout the conception and design. AC, EPB and JAK contributed to the writing of the manuscript. AB provided PPI input throughout the development of the protocol, including the outcomes and background. CH designed the search strategy and performed the preliminary database searches. ES provided clinical input to support the development of the protocol. All authors read and approved the final manuscript.

**Funding** This work was partly funded by the National Institute for Health and Care Research Applied Research Collaboration North West Coast (NIHR ARC NWC). The work was also partly funded by the University of Central Lancashire, Research Institute for Global Health and Wellbeing (grant number: LIFE RSAC3-02).

**Disclaimer** The views expressed are those of the authors and not necessarily those of the NHS, the NIHR, or the Department of Health and Social Care.

**Competing interests** None declared.

**Patient and public involvement** Patients and/or the public were involved in the design, or conduct, or reporting, or dissemination plans of this research. Refer to the Methods section for further details.

**Patient consent for publication** Not applicable.

**Provenance and peer review** Not commissioned; externally peer reviewed.

**ORCID iDs**
Oliver Hamer http://orcid.org/0000-0002-9631-0032
Emma P Bray http://orcid.org/0000-0001-9882-3539

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
