## [Reviewer comments · BMJ Open]

ARTICLE DETAILS

TITLE (PROVISIONAL)	Psychological interventions for weight reduction and sustained weight reduction in adults with overweight and obesity: a scoping review protocol.
AUTHORS	Hamer, Oliver; Bray, Emma; Harris, Cath; Blundell, Amy; Kuroski, Jennifer A; Schneider, Emma; Watkins, Caroline; Clegg, Andrew

VERSION 1 – REVIEW

REVIEWER	Okop, Kufre University of Cape Town
REVIEW RETURNED	20-Jul-2023

GENERAL COMMENTS	This is an important protocol paper focusing on psychological interventions for weight reduction and sustained weight reduction in adults with overweight and obesity. It will add value to the interventions needed to control and manage obesity in many populations. The paper is well written. It is simple and straight to the point. The following are my comments. Abstract: Abstract is well presented The phrase “wholly effective...” would need to be modified. This might mean something different for many categories of readers. I want to believe that you want to indicate that PA and diet interventions implemented so far have not resulted in sustained obesity reduction in the measure expected, particularly in some settings, and or groups of person. Introduction: This is well written. In addition, “wholly effective...” need to be explained or phrased in a way to give a common understanding of ineffectiveness of some PA & diet-related interventions. Lines 33-36: I have not seen the authors mention stress as an important psychological factor. Aim and objectives: This is excellently crafted. However, I will suggest that the authors merge objectives 3 and 5 together, as these are focusing on almost the same outcome/outputs. Methods and Analysis: This is well-ordered. What is the expected outcome and use of this study? It will be of importance to science, to be able to know what exactly the findings of this study (and or its methodology) would add to the body of literature and to science, that might be different from what the many systematics and meta-analysis on obesity interventions have added so far. How would the results be useful to programme implementers, WHO, INGOs, local NGOs, academia, etc?
---

	Authors need to add an appropriate sub-heading to convey this important message to the readers, users - programme managers and researchers, etc. Study limitations and strengths: A brief statement on these might be useful to the readers.
--	--

REVIEWER	Lukes, Sophie University Medical Center of the Johannes Gutenberg University Mainz, Department of Child and Adolescent Psychiatry and Psychotherapy
REVIEW RETURNED	01-Aug-2023

GENERAL COMMENTS	Thank you for the opportunity to review “ Psychological interventions for weight reduction and sustained weight reduction in adults with overweight and obesity: a scoping review protocol.”, whose topic is both interesting and highly relevant. Following, you’ll find some recommendations and concerns: Choice of review method: Please state more clearly why you choose to do a scoping review instead of a systematic review. You mention in the part on strengths and limitations that the review offers to determine gaps in evidence, yet, I think that it would be beneficial not only to name and describe current interventions, but also to see if they have a positive impact on weight loss. Included studies: Do you include studies from a specific period or shall all existing studies on interventions be included? Literature: In the introduction, you name reference 7 (Kyrou I, Christou A, Panagiotakos D, Stefanaki C, Skenderi K, Katsana K, et al. Effects of a hops (Humulus lupulus L.) dry extract supplement on self-reported depression, anxiety and stress levels in apparently healthy young adults: a randomized, placebocontrolled, double-blind, crossover pilot study. Hormones (Athens). 2017;16(2):171-80.) as evidence, that obesity increases the risks for comorbidities, yet, for me, this reference does not seem to be fitting, because, although information on weight is gathered in this study, it focuses on depression and anxiety. Please check – thank you! Outcomes: You mention that information on eating behaviors will be gathered, if available. Please explain in more detail, what you mean with this, does this entail diagnoses of eating disorders, or how are pathological eating behaviors defined? Is it important how this was examined (via interview or questionnaire)? I would recommend to elaborate on this aspect.
---

VERSION 1 – AUTHOR RESPONSE

Reviewer 1	
Abstract is well presented	We have re-phrased the sentence, removing ‘wholly effective’ and instead

The phrase “wholly effective...” would need to be modified. This might mean something different for many categories of readers. I want to believe that you want to indicate that PA and diet interventions implemented so far have not resulted in sustained obesity reduction in the measure expected, particularly in some settings, and or groups of person.	stated that PA and diet interventions have not resulted in sustained obesity reduction (see abstract, page 1): ‘With increasing rates of overweight and obesity, it is evident that a primary focus on diet and exercise has not resulted in sustained obesity reduction within the global population’
Introduction: This is well written. In addition, “wholly effective...” need to be explained or phrased in a way to give a common understanding of ineffectiveness of some PA & diet-related interventions.	In a similar manner to the abstract, we have re-phrased ‘wholly effective’ to state that dietary and physical activity interventions have not resulted in a sustained obesity reduction within the global population (see page 3).
Lines 33-36: I have not seen the authors mention stress as an important psychological factor.	We have added stress as an important factor within line 33, and also highlighted stress within the second paragraph of the introduction (see page 3).
Aim and objectives: This is excellently crafted. However, I have will suggest that the authors merge objectives 3 and 5 together, as these are focusing on almost the same outcome/outputs.	Thank you for the feedback. We are hesitant to merge these two objectives as objective 3 refers specifically to the identification of study characteristics of ‘psychological interventions’, and objective 5 refers to the identification of core characteristics of ‘usual care’.
Methods and Analysis: This is well-ordered. What is the expected outcome and use of this study? It will be of importance to science, to be able to know what exactly the findings of this study (and or its methodology) would add to the body of literature and to science, that might be different from what the many systematics and meta-analysis on obesity interventions have added so far. How would the results be useful to programme implementers, WHO, INGOs, local NGOs, academia, etc? Authors need to add an appropriate sub-heading to convey this important message to the readers, users - programme managers and researchers, etc.	We have included a section on ‘what the findings may add’ towards the end of the methods and analysis section, as suggested. ‘We have stated that the scoping review is likely to identify specific gaps within existing evidence that may be useful in shaping future research initiatives. The identification of these gaps could pave the way for the development of novel innovative psychological interventions and may provide a roadmap for researchers to explore the effectiveness for these interventions. The findings may also assist healthcare providers and key public health organisations gain insight into the types of psychological interventions that can be utilised as a means of weight reduction and sustained weight reduction for adults with overweight and obesity. With a greater understanding of psychological interventions (i.e., the types, characteristics and impact) for weight reduction and sustaining weight reduction, key organisations and healthcare services can tailor

	programs to include evidenced based psychological components' (see page 8).
Study limitations and strengths: A brief statement on these might be useful to the readers.	We have included a 'strengths and limitations' section with bullet points to highlight these to the reader (see page 2), as per the BMJ protocol submission guidance.
Reviewer 2	
Choice of review method: Please state more clearly why you choose to do a scoping review instead of a systematic review.	We have added a further paragraph on page 5 which states why we choose to do a scoping review: 'A scoping review design was chosen because it enables an exploration of the broader spectrum of evidence relating to the range of psychological therapy interventions for adults with overweight and obesity, without being constrained by a rigid inclusion and exclusion criteria of a systematic review. Given the absence of previous reviews which have explored the objectives of this study, the scoping review methodology allows for a more expansive inclusion criteria (e.g., different study designs and intervention types) which will ensure that all relevant studies will be included. The scoping review design has been found to be well-suited for topics that are rapidly evolving and include a heterogeneous body of literature, which is applicable to psychological interventions as a means for achieving weight reduction and sustained weight reduction'.
You mention in the part on strengths and limitations that the review offers to determine gaps in evidence, yet, I think that it would be beneficial not only to name and describe current interventions, but also to see if they have a positive impact on weight loss.	Thank you for the comment, we value the suggestion and agree an effectiveness analysis would be beneficial . That said, we do not plan to include an analysis of effectiveness as this would be more appropriate for a full systematic review rather than a scoping review . However, we fully because we intend to conduct a full systematic review (with meta-analysis) to establish the effectiveness of these interventions once we have the findings from this scoping review. This scoping review will, by its nature, give us a clearer insight into the research available in this area, and will inform the parameters required for a full systematic review. We feel it is important not to miss out this vital

	step. Additionally, the resources we currently have available for this project do not allow us to conduct the full SR at this time. We are unable to conduct an analysis of effectiveness in this scoping review because the project is restricted by time and funding, hence why we have only proposed the scoping element in this protocol.
Included studies: Do you include studies from a specific period or shall all existing studies on interventions be included?	We have clarified in the search strategy that there will be no limiters related to year of publication (see page 5).
Literature: In the introduction, you name reference 7 (Kyrou I, Christou A, Panagiotakos D, Stefanaki C, Skenderi K, Katsana K, et al. Effects of a hops (Humulus lupulus L.) dry extract supplement on self-reported depression, anxiety and stress levels in apparently healthy young adults: a randomized, placebocontrolled, double-blind, crossover pilot study. Hormones (Athens). 2017;16(2):171-80.) as evidence, that obesity increases the risks for comorbidities, yet, for me, this reference does not seem to be fitting, because, although information on weight is gathered in this study, it focuses on depression and anxiety. Please check – thank you!	We have replaced the citation of Kyrou et al, with relevant references from Martin-Rodriguez E, Guillen-Grima F, Martí A, Brugos-Larumbe A. Comorbidity associated with obesity in a large population: The APNA study. Obes Res Clin Pract. 2015 Sep-Oct;9(5):435-47; and Bhaskaran K, Douglas I, Forbes H, dos-Santos-Silva I, Leon DA, Smeeth L. Body-mass index and risk of 22 specific cancers: a population-based cohort study of 5.24 million UK adults. Lancet. 2014 Aug 30;384(9945):755-65. Please see page 3 line 6. Both references highlight the association of overweight and obesity with several key comorbidities (i.e., type 2 diabetes, hypertension etc).
Outcomes: You mention that information on eating behaviours will be gathered, if available. Please explain in more detail, what you mean with this, does this entail diagnoses of eating disorders, or how are pathological eating behaviours defined? Is it important how this was examined (via interview or questionnaire)? I would recommend to elaborate on this aspect.	We have clarified that outcome of eating behaviour will include constructs of dysfunctional, problematic, restrained, emotional, cognitive, and adaptive eating behaviours as measured by instruments such as the Emotional Eating Scale, Intuitive Eating Scale-2, Mindfulness Eating Questionnaire, Dutch Eating Behaviour Questionnaire, Three-Factor Eating Questionnaire, etc (see page 7).

VERSION 2 – REVIEW

REVIEWER	Okop, Kufre University of Cape Town
REVIEW RETURNED	18-Sep-2023

GENERAL COMMENTS	The author has made all needed revisions, and the manuscript as it stands, is in good form for publication.
---

REVIEWER	Lukes, Sophie University Medical Center of the Johannes Gutenberg University Mainz, Department of Child and Adolescent Psychiatry and Psychotherapy
REVIEW RETURNED	04-Oct-2023

GENERAL COMMENTS	Thank you for the revised edition of this interesting manuscript. All my previous aspects and questions were sufficiently answered and revised. I have just one small last recommendation for the introduction: I would recommend to restructure the sentence „While these factors are critical for weight loss, they ignore the complex psychological (e.g., fears), emotional (e.g., stress), and behavioural factors (e.g., avoidance) that underpin overweight and obesity (6, 15-18).“ to „While these factors are critical for weight loss, they ignore the complex psychological factors (e.g., emotional, such as fears and behavioural, such as avoidance) that underpin overweight and obesity (6, 15-18).“ since both emotional and behavioural factors are psychological factors and this sentence structure could be potentially misunderstanding to some readers. Thank you.
--

VERSION 2 – AUTHOR RESPONSE

Thank for the feedback. We have re-upload the supplementary file in PDF format and have restructured the sentence as per reviewer two's suggestion: I would recommend to restructure the sentence 'While these factors are critical for weight loss, they ignore the complex psychological (e.g., fears), emotional (e.g., stress), and behavioural factors (e.g., avoidance) that underpin overweight and obesity (6, 15-18).' to 'While these factors are critical for weight loss, they ignore the complex psychological factors (e.g., emotional, such as fears and behavioural, such as avoidance) that underpin overweight and obesity (6, 15-18).'